# Long-Term Effectiveness of BNT162b2 Pfizer-BioNTech mRNA-Based Vaccine on B Cell Compartment: Efficient Recall of SARS-CoV-2-Specific Memory B Cells

**DOI:** 10.3390/ijms232315046

**Published:** 2022-11-30

**Authors:** Rosalia Busà, Monica Miele, Maria Concetta Sorrentino, Giandomenico Amico, Francesca Timoneri, Vitale Miceli, Mariangela Di Bella, Giovanna Russelli, Alessia Gallo, Giovanni Zito, Gioacchin Iannolo, Pier Giulio Conaldi, Matteo Bulati

**Affiliations:** 1Research Department, Mediterranean Institute for Transplantation and Advanced Specialized Therapies (IRCCS ISMETT), 90127 Palermo, Italy; 2Ri.MED Foundation, 90133 Palermo, Italy; 3Department of Laboratory Medicine and Advanced Biotechnologies, Mediterranean Institute for Transplantation and Advanced Specialized Therapies (IRCCS ISMETT), 90127 Palermo, Italy

**Keywords:** SARS-CoV-2, mRNA vaccine, BNT162b2, booster dose, B cells, long-lasting memory

## Abstract

At present, there is a lack of clinical evidence about the impact and long-term durability of the immune response induced by the third dose of mRNA vaccines. In this study, we followed up the B cell compartment behavior in a cohort of immunocompetent individuals three and six months after the third dose of vaccine. During this period, some subjects contracted the virus. In uninfected vaccinated subjects, we did not report any changes in serum spike-specific IgG levels, with a significant reduction in IgA. Instead, subjects recovered from natural infection showed a significant increase in both specific IgG and IgA. Moreover, we showed a time-related decrease in IgG neutralizing potential to all SARS-CoV-2 variants of concern (VOC) in uninfected compared to recovered subjects, who displayed an increased neutralizing ability, particularly against the omicron variant. Finally, we underlined the presence of a pool of SARS-CoV-2-specific B cells in both groups that are prone to respond to restimulation, as demonstrated by their ability to differentiate into plasma cells and to produce anti-SARS-CoV-2-specific immunoglobulins. These data lead us to assert the long-term effectiveness of the BNT162b2 vaccine in contrasting the severe form of the pathology and prevent COVID-19-associated hospitalization.

## 1. Introduction

Vaccination is the main “weapon” in the global fight against the COVID-19 pandemic. The protective role of the mRNA-based SARS-CoV-2 BNT162b2 Pfizer/BioNTech vaccine has been asserted in several studies [1,2,3,4,5]. During the pandemic, the rapidly increasing number of COVID-19 cases among previously vaccinated individuals led to the administration of an additional third dose of vaccine as a booster [6,7]. Many authors have described an immediate improvement in immune response after the booster dose [8,9,10,11,12]. At present, there is a lack of clinical evidence about the impact and long-term durability of the immune response induced by the third dose of mRNA vaccines. In a previous study, we longitudinally analyzed the antibody- and SARS-CoV-2-specific memory B-cell responses for 9 months after 2 primary doses of the BNT162b2 Pfizer-BioNTech vaccination and 3 weeks after the third dose in a cohort of 11 immunocompetent, uninfected, vaccinated individuals [9]. Although anti-spike IgG and IgA antibodies response had decreased over nine months, all subjects included in the study had maintained SARS-CoV-2 memory B cells, suggesting that the mRNA-based vaccine can induce persistent immune memory after the second dose of vaccination. Moreover, an additional third booster dose brings both SARS-CoV2-specific antibodies and memory B cells back up to high levels [9]. In this studied, conducted an in-depth analysis of the B-cell compartment of the same immunocompetent cohort three and six months after the third dose of the BNT162b2 vaccine. During the follow-up between four and five months after the third dose, 5/11 subjects contracted the virus and developed mild symptoms, such as fever, pain, sore throat, chills, and cough. We evaluated the anti-spike IgG and IgA from T0 (three weeks after the second dose) to T4 (six months after the third dose), as well as the neutralizing activity against variants of concern (VOCs) at T2 (three weeks after the third dose), T3 (three months after the third dose), and T4. Moreover, we assessed the functionality of SARS-CoV-2-specific B cells by performing an in vitro polyclonal activation of the PBMCs at T3 and T4. Finally, we evaluated circulating SARS-CoV-2-specific B cells, focusing on their subpopulations (naïve/memory and plasma blasts). In the study cohort, we observed a gradual reduction in immunoglobulin secretion, neutralizing capacity, and SARS-CoV-2-specific B-cell functionality at T3, whereas at T4, uninfected vaccinated individuals exhibited the same trend of reduction in SARS-CoV-2-specific response, which, in contrast, was increased in recovered subjects. SARS-CoV-2 B cells analyzed at T4 revealed that, in contrast to uninfected vaccinated individuals, in recovered vaccinated subjects, there was a significant increase in circulating SARS-CoV-2-specific plasma blasts, which rapidly produce anti-spike IgG and IgA antibodies, as previously described [13]. Furthermore, SARS-CoV-2-specific IgG produced by recovered vaccinated subjects was associated with an improved neutralizing ability against all VOCs, including omicron, compared to uninfected vaccinated individuals. In contrast to the time-related, post-vaccination decline in antibodies, SARS-CoV-2-specific memory B cells appeared to be highly stable over time. Upon re-exposure to antigens, either through in vitro stimulation or natural infection, these memory B cells differentiate into antibody-secreting cells and rapidly produce highly functional antibodies.

## 2. Results

### 2.1. SARS-CoV-2-Specific IgG and IgA Kinetics after Pfizer-BioNTech BNT162b2 mRNA Vaccine

Figure 1A shows the timing of sampling for the evaluation of the humoral response (anti-SARS-CoV-2-specific IgG and IgA) to the vaccine in our studied cohort. Figure 1 depicts the IgG (Figure 1B) and IgA (Figure 1C) kinetics during the follow-up study. We previously described [9] that three weeks after the second dose of vaccine (T0), the median value of specific IgG was 657.8 BAU/mL (SEM = 78.59), which significantly decreased (*p* = 0.0009) to 209 BAU/mL (SEM = 42.86) in the next nine months (T1). Three weeks after the third dose (T2), we observed a significant increase in specific IgG (Median = 4134 BAU/mL, SEM = 430) compared to both T0 (*p* = 0.0038) and T1 (*p* = 0.0020), demonstrating the efficacy of the third dose to boost antibody response against SARS-CoV-2. To estimate the persistence of the SARS-CoV-2-specific humoral response induced by a booster dose, we assessed the measurement of serum IgG at T3 and T4. Unlike after the second dose, both three (T3) and six months (T4) after the third dose, there was no significant reduction in SARS-CoV-2-specific IgG. At T3, we observed a slight decrease in the median value of IgG, which was 3354 BAU/mL (SEM = 568.7). Starting from the T3 time point, 5/11 subjects contracted SARS-CoV-2 infection; accordingly, the analysis has was differentiated between vaccinated and hybrid-immunized subjects and vaccinated subjects who contracted the infection. At T4, we observed an insignificant increase in specific IgG (median= 4030 BAU/mL, SEM = 641.9) compared to T3. However, to better evaluate the humoral responses of SARS-CoV-2-negative and SARS-CoV-2-positive subjects, we compared the IgG levels between the groups. As shown in Figure 1D, at T4, the median value of SARS-CoV-2-specific IgG was significantly higher (*p* = 0.0229) in recovered subjects (6000 BAU/mL, SEM = 428.7) compared to uninfected subjects (1581 BAU/mL, SEM = 562.7). Concerning SARS-CoV-2-specific IgA, we showed comparable kinetics to IgG only up to T2. As shown in Figure 1C, serum IgA had a 9.47 median ratio (SEM = 1.277) at T0, which significantly decreased up to 1.100 (SEM = 0.3285) at T1 (*p* = 0.0006), with a significant upsurge (*p* < 0.0001) after the booster dose (T2) compared to T1, reaching a median ratio of 9.80 (SEM = 0.5248). At T3, we observed a drastic drop-off in anti-spike IgA serum levels (median = 1.00, SEM = 0.090) compared to T2, which decreased further at T4 (median = 0.87, SEM = 0.914). In Figure 1E, we separated the two groups of vaccinated at T4, observing that vaccinated subjects who had contracted the infection showed significantly higher levels of serum IgA (median = 5.840, SEM = 0.557; *p* = 0.0043) compared to uninfected subjects (median = 0.3850, SEM = 0.1157). These data suggest that the humoral response induced by mRNA vaccines declines over time and that booster doses can restimulate the humoral response similarly to natural infection.

### 2.2. SARS-CoV-2-Specific IgG Neutralization Ability against SARS-CoV-2 Variants

To establish the quality of antibody responses, we assessed the IgG neutralizing potential in all SARS-CoV-2 VOCs (wild type, alfa UK, beta, gamma, delta, and omicron) using a ProcartaPlex human SARS-CoV-2 variants neutralizing antibody assay (Thermo Fisher Scientific, Waltham, MA, USA) after the third dose or at T2, T3, and T4. As shown in Figure 2, we observed an insignificant reduction in SARS-CoV-2-specific IgG neutralization ability for all variants between T2 and T3 (3 months after the third dose). On the contrary, we reported a significant reduction in SARS-CoV-2-specific IgG neutralization ability only in vaccinated subjects between T2 and T4 (6 months after the third dose) against the SARS-CoV-2 wild type (median = 99%, SEM = 0.975 at T2; median = 86.30%, SEM = 3.350 at T4; *p* = 0.018) (Figure 2A), alfa UK (median = 99.25%, SEM = 2.140 at T2; median = 65.87%, SEM = 5.574 at T4; *p* = 0.003) (Figure 2B), beta (median = 87.9%, SEM = 4.672 at T2; median = 61.51%, SEM = 3.722 at T4; *p* = 0.042) (Figure 2C), gamma (median = 87.1%, SEM = 5.014 at T2; median = 60.46%, SEM = 3.99 at T4; *p* = 0.029) (Figure 2D), and delta (median = 98.2%, SEM = 2.675 at T2; median = 66.95%, SEM = 4.164 at T4; *p* = 0.010) (Figure 2E) VOCs. At both time points, the median of the percentages of neutralization of IgG against the omicron variant was the lowest in comparison to the other VOCs. At T2, the median percentage of neutralization was 41.38% (SEM = 10.24), with a decrease to 38.16% at T4 (SEM = 9.25) in vaccinated subjects only (Figure 2F). Moreover, a comparison of the IgG neutralizing ability of uninfected vaccinated versus recovered vaccinated subjects revealed a significant increase in IgG neutralizing capacity against all VOCs (wild type (*p* = 0.007), alfa UK (*p* = 0.001), beta (*p* = 0.007), gamma (*p* = 0.007), delta (*p* = 0.003), and omicron (*p* = 0.028).

Interestingly, normalizing the SARS-CoV-2-specific IgG neutralization of VOCs vs. the wild type, we observed a general reduction in neutralization ability versus all VOCs in never-infected subjects (Figure 3A); in particular, we observed a significant decrease in the gamma (*p* = 0.0332) and omicron (*p* < 0.0001) VOCs. With respect to recovered subjects (Figure 3B), we observed only a slight and insignificant reduction in the omicron VOC. Finally, to better quantify the increased index in neutralization ability in recovered subjects, we compared the fold change of the two groups (Figure 3C), finding a significant increase in neutralization ability in recovered subjects compared to uninfected subjects against the beta (*p* = 0.0363), gamma (*p* = 0.0057), and omicron (*p* < 0.0001) variants, in which we observed a twofold increase.

### 2.3. The SARS-CoV-2-Specific B-Cell Pool Is Stable until Six Months after the Third Dose of the Pfizer-BioNTech BNT162b2 mRNA Vaccine, Preserving Its Effector Function

SARS-CoV-2-specific B cells were characterized using a spike tetramer solution by combining the SARS-CoV-2-biotinylated-recombinant protein and two distinct fluorescently labelled streptavidin conjugates. SARS-CoV-2-specific B cells were evaluated by flow cytometry to detect the expression of cell-surface IgG, IgA, and IgM isotypes at T1, T2, and T4. We previously demonstrated the presence of a pool of SARS-CoV-2-specific B cells (median = 0.49%, SEM = 0.061) nine months after the second dose (T1), which significantly increased (*p* = 0.024) three weeks after the third dose (T2) (median = 0.81%, SEM = 0.139) [9]. As depicted in Figure 4A, at T4, there is a significant decrease (*p* = 0.006) in SARS-CoV-2-specific B cells (median = 0.40%, SEM = 0.046) compared to T2. A comparison of the percentage of SARS-CoV-2-specific B cells at T1 and T4 does not reveal any difference. At T4, we did not observe any difference in the percentage of SARS-CoV-2-specific B cells between uninfected and recovered vaccinated subjects (Figure A1A), as previously reported [14]. Analysis of immunoglobulin surface expression of spike-specific B cells reveals that most of these cells express IgG (median = 72.73%, SEM = 2.380 at T1; median = 77.46%, SEM = 2.656 at T2; median = 61.33%, SEM = 4.688 at T3) (Figure 4B), and the remaining cells express IgA (median = 6.29%, SEM = 0.911 at T1; median = 7.37%, SEM = 1.101 at T2; median = 3.04%, SEM = 0.680 at T3) (Figure 4C) or IgM (median = 15.04%, SEM = 2.837 at T1; median = 12.30%, SEM = 2.590 at T2; median = 24.97%, SEM = 4.465 at T3) (Figure 4D) without any significant differences between T1 and T2. A comparison of T4 to both T1 and T2 reveals a significant reduction in IgG (*p* = 0.002 T1 vs. T4; *p* < 0.0001 T2 vs. T4) or IgA (*p* = 0.023 T1 vs. T4; *p* = 0.014 T2 vs. T4) expression, accompanied by a significant increase in IgM expression (*p* = 0.014 T1 vs. T4; *p* = 0.001 T2 vs. T4) on SARS-CoV-2-specific B cells. At T4, we did not observe any significant differential expression of IgG, IgA, or IgM on SARS-CoV-2-specific B cells between uninfected and recovered subjects (Figure A1B–D). Figure A2 shows the gating strategy and a representative flow cytometry analysis that we used to analyze immunoglobulin isotype expressions on spike-specific B cells. Briefly, CD19^+^ total B cells were gated on double-positive streptavidin conjugates for the quantification of spike-specific B cells. Subsequently, the surface expression of IgG, IgA, and IgM immunoglobulin was quantified. Moreover, to verify the reactivity against the virus of SARS-CoV-2-specific B cells, we performed an in vitro polyclonal activation of the PBMCs from all subjects enrolled in the study at T3 and T4. As depicted in Figure 4E, the activation of donor PBMCs was successful, as we observed SARS-CoV-2-specific IgG production responses at both time points, without any significant difference (*p* = 0.640). These results indicate that the pool of SARS-CoV-2-specific B cells preserves its effector function. As expected, at T4 (Figure 4F), recovered subjects had an increased but not significant ability to produce anti-spike IgG relative to their uninfected counterparts (*p* = 0.229). Figure 4G shows a representative FluoroSpot assay in which stimulated and unstimulated PBMCs from each subject were seeded on a precoated anti-IgG FluoroSpot plate for antigen-specific analysis, with total IgG spots as a positive control.

### 2.4. SARS-CoV-2-Specific B-Cell Subpopulations Are Enriched in Plasma Blasts in Recovered Subjects

Based on the obtained results, which show a more efficient B-cell response in recovered subjects, we analyzed the differentiation stages among SARS-CoV-2-specific B-cells at T4 in both uninfected and recovered subjects. As shown in Figure 5A, we did not report any differences between the two groups of subjects in terms of memory/naïve distribution using IgD and CD27 markers. In both groups, memory-switched (IgD^−^CD27^+^) cells are the main population represented (median = 59.92%, SEM = 7.688 in uninfected; median = 68.33%, SEM = 2.556 in recovered), followed by naïve (IgD^+^CD27^−^) (median = 17.65%, SEM = 8.073 in uninfected; median = 20.82%, SEM = 2.659 in recovered), memory-unswitched (IgD^+^CD27^+^) (median = 7.44%, SEM = 3.285 in uninfected; median = 7.98%, SEM = 0.801 in recovered), and late-memory (IgD^−^CD27^−^) (median = 2.98%, SEM = 1.785 in uninfected; median = 3.25%, SEM = 0.722 in recovered) B cells. Analysis of SARS-CoV-2-specific B cells using CD27 and CD38 markers, revealed a significant increase (*p* = 0.043) in CD27^high^CD38^high^ plasma blasts in recovered (median = 7.5%, SEM = 1.389) compared to uninfected subjects (median = 1.3%, SEM = 0.540) (Figure 5B). Figure A2 shows a representative flow cytometry analysis used to analyze spike-specific B-cell subpopulations. Briefly, CD19^+^ total B cells were gated on double-positive streptavidin conjugates. On this gate of spike-specific B cells, we quantified IgD, CD27, and CD38 surface expressions.

## 3. Discussion

To preserve immunity at protective levels, the quality and persistence of the immune response elicited by infection or vaccination must be defined. To achieve this, it is necessary to determine the durability and quality of protection offered by infection and/or vaccination and explore the immune mechanisms related to safeguarding against severe COVID-19 disease. Virus-specific memory B and T cells have been established to play a pivotal role in SARS-CoV-2 immunity [15,16,17], although their protective contribution is less clearly defined. mRNA-based SARS-CoV-2 vaccination engenders protective immunity against the virus by inducing strong antibody responses and long-lasting memory B cells that promptly respond and produce new antibodies upon antigen re-exposure [9]. However, it remains unclear how three doses of vaccination affect the magnitude and quality of immune responses, particularly against immune-evasive SARS-CoV-2 variants such as omicron in the prevention of COVID-19-associated hospitalization and severe disease and, finally, how long this protection lasts. In this study, to assess the long-term effectiveness of the BNT162b2 Pfizer-BioNTech mRNA-based vaccine, we followed-up on humoral response and the B-cell compartment behavior in a cohort of IC subjects three and six months after the third dose of the vaccine. We reported that, in contrast to observations after the second dose [9], both three and six months after the third dose of vaccine, we did not observe a significant reduction in serum spike-specific IgG levels. Additionally, recovered subjects displayed significantly higher levels of serum SARS-CoV-2-specific IgG compared to uninfected subjects. Conversely, serum spike-specific IgA waned over time, up to negativization, even after the third dose of vaccine, except for subjects who recovered from natural infection, with a significant upsurge of serum IgA. We also observed the same trend regarding the quality of antibody response, which showed a time-related decrease in IgG neutralizing potential to all SARS-CoV-2 VOCs in uninfected subjects, as opposed to recovered subjects, who displayed an increased neutralizing ability, particularly against the omicron variant. These results are in line with the current view that hybrid immunity (vaccination plus SARS-CoV-2 infection in any order) offers greater protection than immunity elicited by vaccination or COVID-19 separately [16,18,19,20,21]. Concerning SARS-CoV-2-specific B cells, we can assume the existence of a pool of memory B cells that are prone to respond to restimulation by both in vitro stimulation and natural infection, as demonstrated by the ability of these cells to differentiate into plasma cells and to produce anti-SARS-CoV-2-specific immunoglobulins. These potentially functional antigen-specific reactivation responses seem to be robust, with little to no waning over time, as previously shown [22]. Finally, we observed an increase in SARS-CoV-2 IgM^+^ specific B cells at six months (T4). This is a surprising result, as we expected the persistence of IgG^+^ memory B cells among the IgG/IgA/IgM isotypes. At T4, we did not report any significant differential expression of IgG, IgA, and IgM on SARS-CoV-2-specific B cells between uninfected and recovered subjects. Although not significant, we observed a trend of increased IgM isotypes in the former compared to the latter. To explain this interesting result, recent evidence seems to support a protective role of IgM^+^ memory B cells against SARS-CoV-2, along with other coronaviruses, as outlined by the worst clinical outcomes observed in COVID-19 patients with impaired IgM^+^ memory response [23]. This peculiar subset of memory B cells, which probably originates outside germinal centers in a T-cell-independent pathway, provides a rapid line of defense against mucosal infections [24]. This is an interesting result, owing to the urgent need for immunological markers to better characterize both COVID19 pathogenesis and its evolution or sequelae, as well as vaccine effectiveness. For this purpose, among other useful immunological markers, such as SARS-CoV-2-specific IgG or IgA, SARS-CoV-2-specific IgM^+^ memory B cells stand out as probable markers of clinical outcome or vaccine efficacy. Altogether, these data lead us to assert the long-term effectiveness of the BNT162b2 Pfizer-BioNTech mRNA vaccine in contrasting the severe form of the pathology and preventing COVID-19-associated hospitalization. A limitation of our study is the restricted cohort of subjects involved; this represents a key point, given that the recruited subjects are a group of healthy healthcare workers with the same vaccination history and a close-contact work environment. In conclusion, understanding and measuring the individual persistence of immune protection is important for the management and the control of the pandemic, as well as with respect to future vaccination campaigns to contrast the risk of emerging novel viral variants.

## 4. Materials and Methods

### 4.1. Subjects Enrolled in the Study

At the time of SARS-CoV-2 vaccination with the Pfizer-BioNTech BNT162b2 mRNA vaccine, 11 immunocompetent (ICs) healthy subjects (5 male and 6 female; median age, 44, range 33–51) were enrolled. Enrolled subjects never had positive nasopharyngeal swabs (NPS) and anti-N response until three months after the third dose, whereas 5/11 contracted infection (omicron variant) between four and five months after the third dose. Blood, PBMCs, and serum samples were collected three weeks (T0) and nine months (T1) after the second dose and three weeks (T2), three months (T3), and six months (T4) after the third booster dose for the analysis of spike-specific humoral and B-cell immune responses. Blood was sampled from subjects who contracted the disease one month after negativization, corresponding to approximately six months after the third dose (T4). This study was approved by the IRCCS-ISMETT Institutional Research Review Board (IRRB 00/21) and by the Ethics Committee of ISMETT; all enrolled individuals signed a written informed consent form.

### 4.2. Detection of SARS-CoV-2 Antibodies

LIAISON^®^ SARS-CoV-2 S1/S2 IgG chemiluminescent immunoassays (CLIA) (DiaSorin S.p.A. Saluggia, VC, Italy) were used on a fully automated LIAISON^®^ XL analyzer (DiaSorin S.p.A., Saluggia (VC), Italy) to detect anti-spike IgG from serum samples. The concentration of SARS-CoV-2 S1/S2 IgG was expressed as binding antibody units (BAU/mL), and values > 33.8 BAU/mL were considered positive. An anti-SARS-CoV-2 IgA enzyme-linked immunoassays (ELISA) (EUROIMMUN, Perkin Elmer Company, Lübeck, Germany) were used on a fully automated EUROIMMUN Analyzer I (EUROIMMUN, Perkin Elmer Company, Lübeck, Germany) for semi-quantitative detection of IgA to S1 fragments of the virus. Anti-SARS-CoV-2 IgA concentrations were expressed as a ratio of the extinction of the sample to that of the calibrator (OD value of serum sample/OD value of the calibrator), and a ratio > 1.1 was considered positive.

### 4.3. Neutralization Assay

SARS-CoV2 neutralizing antibodies were detected using a ProcartaPlex Human SARS-CoV-2 variant neutralizing antibody assay (Thermo Fisher Scientific, Massachusetts, USA) according to the manufacturer’s instructions. The level of neutralization was determined using Luminex™ magnetic bead technology. This assay is designed to simultaneously detect the neutralizing potential of antibodies towards both wild type (WT) and five described variants (alfa UK, beta, gamma, delta, and omicron). Briefly, this kit contains specific capture beads coupled with spike S1 protein (RBD) from the above-described variants. Samples with neutralizing antibodies compete with an excess amount of biotinylated ACE2, which binds to the proteins on the beads and produces a fluorescence signal. Signals are inversely proportional to the level of neutralizing antibodies, as this is a competitive immunoassay. Each kind of neutralizing antibody was quantified using a Luminex 200 instrument, which utilizes xMAP technology with multiple-analyte profiling and xPONENT 4.2 software (Luminex Corp., Austin, TX, USA). The neutralization (%) for each sample was calculated based on the median fluorescence intensity (MFI) of the sample and the MFI of the negative control (1 − (MFI of samples/MFI of negative control) × 100). The cutoff level was determined to be 20%. 

### 4.4. Isolation and Quantification of SARS-CoV-2-Specific B Cells

Peripheral blood mononuclear cells (PBMCs) were isolated from venous blood by density gradient centrifugation on Lympholyte cell separation media (Cedarlane Laboratories Limited, Burlington, ON, Canada). Subsequently, CD19^+^ B cells were isolated from PBMCs using anti-CD19 magnetic microbeads (REAlease CD19 MicroBeads Kit, Miltenyi Biotec, Auburn, CA, USA). The CD19^+^ obtained B cells showed a purity yield higher than 98%, as determined by flow cytometry analysis. The isolated fraction was stained with a SARS-CoV-2 spike B-cell analysis kit (Miltenyi Biotec, Auburn, CA, USA), following the manufacturer’s instructions, and with anti-human CD19_BV786_,CD27_BV605_, CD38_PE-CF594_, IgD_FITC_ (Becton Dickinson, San Jose, CA, USA), IgM_APC_, IgG_BV421_, and IgA_BV510_ (Miltenyi Biotec, Auburn, CA, USA) monoclonal antibodies to quantify the SARS-CoV-2-specific B cells and their related subpopulations at T4. Samples were analyzed using a 16-colors FACS Celesta SORP flow cytometer (Becton Dickinson, San Jose, CA, USA) with the same instrument setting. At least 10^4^ B cells were analyzed using FlowJo™ v10.8.1 software (BD Life Sciences, Oxford, UK).

### 4.5. IgG SARS-CoV-2 FluoroSpot Assay

SARS-CoV-2-specific memory B cells were evaluated in the cryopreserved PBMCs using polyclonal stimulation in RPMI 1640 + 10% FCS (complete medium) in the presence of R848 (1 µg/mL) and IL-2 (10 ng/mL) at a cell density of 10^6^ PBMCs/mL for 3 days. Unstimulated PBMCs were cultured in a complete medium supplemented with IL-2 (10 ng/mL) for 3 days. B cells secreting IgG antibodies (ASC) specific to the SARS-CoV-2 spike protein were enumerated using the human IgG SARS-CoV-2 (Spike) Fluorospot Path kit (Mabtech AB, Cincinnati, OH, USA). Briefly, stimulated and unstimulated PBMCs (350,000 per well) were washed and seeded in duplicate in a complete medium on a precoated anti-IgG FluoroSpot plate for antigen-specific analysis. Then, 50,000 cells were seeded for total IgG spots as a positive control. The number of ASCs specific to spike protein and cells secreting IgG (total IgG) spots were detected according to the manufacturer’s protocol (Mabtech AB). ASC spots were measured on an AID vSpot Spectrum Elispot/Fluorospot reader system using AID Elispot software version 7.x. ASC counts were normalized to ASCs per million PBMCs for all analyses after subtracting the background spots of the negative control (unstimulated cells and pre-COVID pandemic PBMC sample).

### 4.6. Statistical Analysis

Graph Pad Prism 9.0 (Graph Pad Software, San Diego, CA, USA) software was used to perform statistical analysis. Depending the type of samples being compared, Wilcoxon matched-pairs non-parametric test, Dunnett’s test, Mann–Whitney test, or one-way ANOVA tests with multiple comparisons were used. *p* < 0.05 was considered significant. 

## Figures and Tables

**Figure 1 ijms-23-15046-f001:**
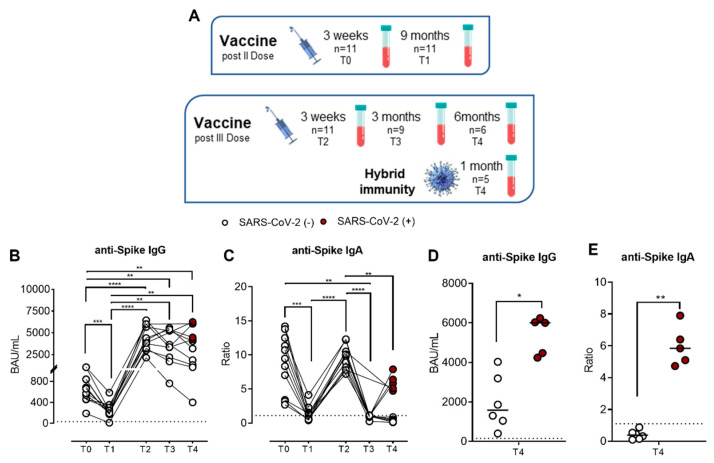
Kinetics of total anti-SARS-CoV-2 IgG and IgA serum antibody levels (n = 11) of the Pfizer-BioNTech BNT162b2 mRNA-based vaccination in immunocompetent (IC), healthy subjects. (**A**) Schematic timing of sampling for the evaluation of the humoral response to the vaccine in our studied cohort. Both serum antibodies were evaluated three weeks (T0) and nine months (T1) after the second dose and three weeks (T2), three months (T3), and six months (T4) after the third booster dose for uninfected subjects or one month after a SARS-CoV-2-negative swab for infected subjects. Vaccinated subjects who contracted SARS-CoV-2 infection during the follow-up (n = 5) are indicated as SARS-CoV-2 (+) (red dots), whereas uninfected subjects (n = 6) are as SARS-CoV-2 (−) (white dots). (**B**) Anti-SARS-CoV-2 S1/S2 IgG levels and (**C**) anti-SARS-CoV-2 S1 IgA levels at T0, T1, T2, T3, and T4. (**D**) Comparison of anti-SARS-CoV-2 S1/S2 IgG levels and (**E**) anti-SARS-CoV-2 S1 IgA levels at T4 between vaccinated SARS-CoV-2 (+) and SARS-CoV-2 (−) subjects (n = 6). The dotted lines correspond to IgG (>33.8 BAU/mL) and IgA (>1.1 Ratio) cutoff, respectively. The significance was determined using Tukey’s multiple comparisons test, one-way ANOVA, Mann–Whitney test, and *t*-tests; * *p* < 0.0332; ** *p* < 0.0021; *** *p* < 0.0002; **** *p* < 0.0001.

**Figure 2 ijms-23-15046-f002:**
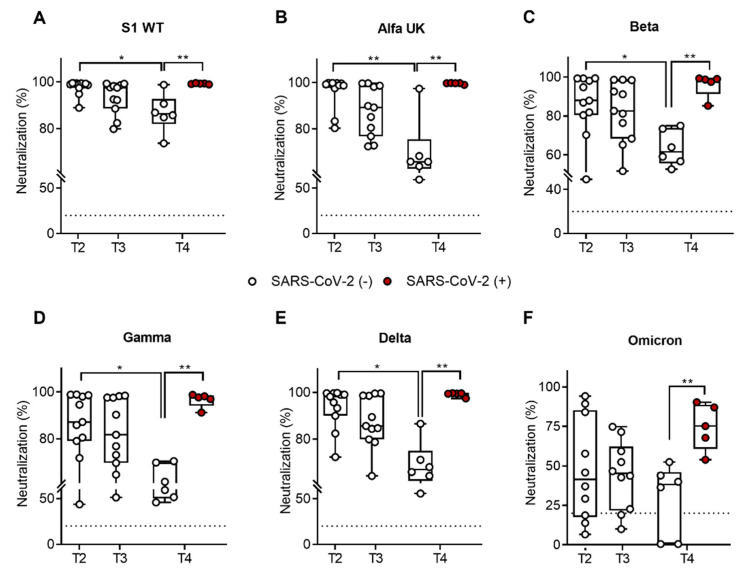
Percentage (%) of neutralizing activity to all SARS-CoV-2 VOCs in immunocompetent (ICs) healthy subjects (n = 11) after the Pfizer-BioNTech BNT162b2 mRNA-based vaccination. IgG neutralizing potential was evaluated three weeks (T2), as well as three (T3) and six months (T4), after the third booster dose. The time point T4 includes vaccinated subjects who contracted SARS-CoV-2 infection during the follow-up (n = 5), indicated as SARS-CoV-2 (+) (red dots), whereas uninfected subjects (n = 6) are indicated as SARS-CoV-2 (−) (white dots). The SARS-CoV-2-specific IgG neutralization ability against the SARS-CoV-2 wild type (**A**), alfa UK (**B**), beta (**C**), gamma (**D**), delta (**E**), and omicron (**F**). The dotted lines correspond to the cutoff level (>20%). The significance was determined using the Mann–Whitney test and *t*-tests; * *p* < 0.0332; ** *p* < 0.0021.

**Figure 3 ijms-23-15046-f003:**
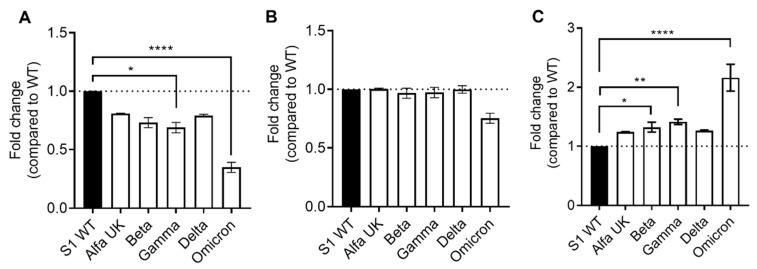
Fold change in SARS-CoV-2-specific IgG neutralization against VOCs (alfa UK, beta, gamma, delta, and omicron) normalized versus the wild type (WT) in uninfected (**A**) and recovered (**B**) vaccinated subjects at T4. (**C**) Comparison of fold change between the two analyzed groups at T4. Significance was determined using Dunnett’s multiple comparisons test; * *p* < 0.0332, ** *p* < 0.0021, **** *p* < 0.0001.

**Figure 4 ijms-23-15046-f004:**
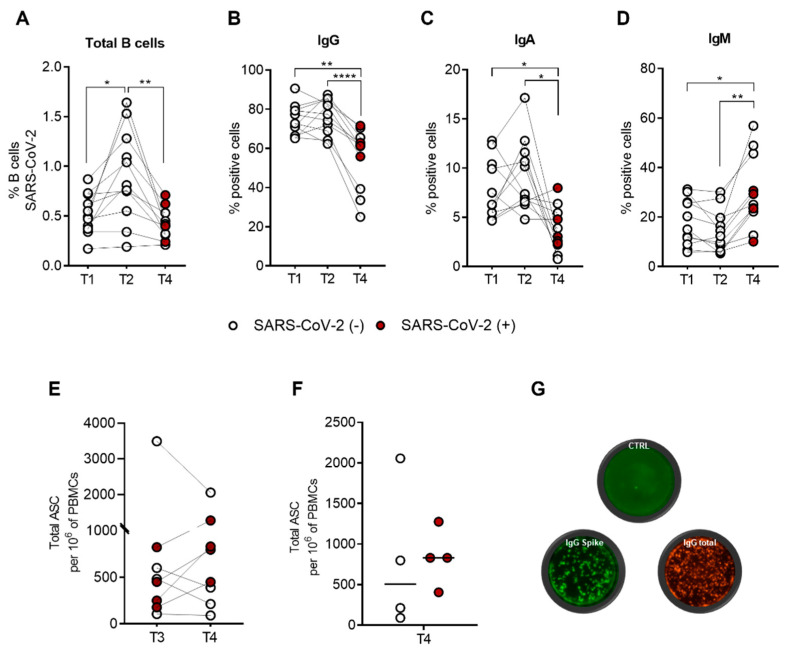
SARS-CoV-2-specific memory B-cell response in immunocompetent (IC), healthy subjects (n = 11) following Pfizer-BioNTech BNT162b2 mRNA-based vaccination. Spike-specific B cells were identified by flow cytometry nine months (T1) after the second dose and three weeks (T2) and six months (T4) after the third booster dose. Time point T4 includes vaccinated subjects who contracted SARS-CoV-2 infection during the follow-up (n = 5), indicated as SARS-CoV-2 (+) (red dots), and the uninfected (n = 6) are indicated as SARS-CoV-2 (−) (white dots). For recovered subjects, time point T4 corresponds to one month after a SARS-CoV-2-negative swab. (**A**) Percentage (%) of SARS-CoV-2-specific B cells at T1, T2, and T4. Comparison of percentage (%) of positive cell to surface immunoglobulin isotypes, IgG (**B**), IgA (**C**), and IgM (**D**) at T1, T2, and T4. (**E**) Total B-cells secreting IgG antibodies (ASC) specific to the SARS-CoV-2 spike protein at times T3 and T4. (**F**) Comparison between vaccinated SARS-CoV-2 (+) (n = 5) and SARS-CoV-2 (−) (n = 6) only at T4. (**G**) Representative examples of single-subject anti-IgG FluoroSpot assays. The control well represents unstimulated PBMCs, the IgG spike well is the stimulated PBMCs, and the IgG total well is the positive control. Significance was determined using Tukey’s multiple comparisons tests, one-way ANOVA, Wilcoxon matched-pairs test, Mann–Whitney test, and *t*-tests; * *p* < 0.0332: ** *p* < 0.0021; **** *p* < 0.0001.

**Figure 5 ijms-23-15046-f005:**
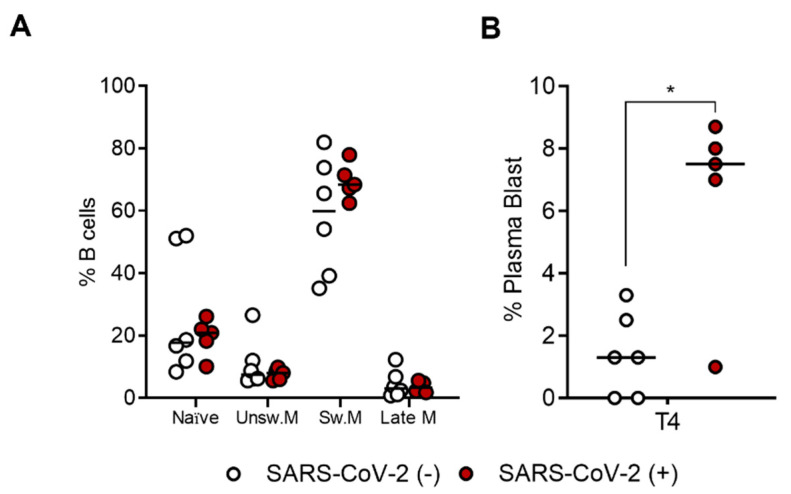
Comparison of SARS-CoV-2-specific B-cell subpopulations between recovered SARS-CoV-2 (+) (n = 5) and uninfected SARS-CoV-2 (−) (n = 6) vaccinated subjects at T4. (**A**) Percentage (%) of each SARS-CoV-2-specific B-cell subpopulations (naïve, IgD^+^CD27^−^; unswitched memory, IgD^+^CD27^+^; switched memory, IgD^−^CD27^+^; and late memory, IgD^−^CD27^−^) at T4 in both uninfected (white dots) and recovered subjects (red dots). (**B**) Percentage (%) of plasma blast (CD27^high^CD38^high^) in both uninfected SARS-CoV-2 (−) and recovered SARS-CoV-2 (+) subjects at T4. Significance was determined using the Mann–Whitney test and *t*-tests, * *p* < 0.0332.

## Data Availability

The raw data supporting the conclusions of this article will be made available by the authors, without undue reservation, to any qualified researcher.

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
