# Peer review of "Long-Term Effectiveness of BNT162b2 Pfizer-BioNTech mRNA-Based Vaccine on B Cell Compartment: Efficient Recall of SARS-CoV-2-Specific Memory B Cells"

_ijms, 2022, doi:10.3390/ijms232315046_

Round 1

Reviewer 1 Report

The manuscript by Rosalia Busà et al. reports the results of a study that evaluated functionality of B cells, including different SARS-CoV-2-specific B cell subpopulations, in a cohort of immunocompetent individuals at three and six months from the third dose of mRNA vaccine. During the follow up period some of the individuals contracted the virus. The authors compared time-related serum Spike-specific IgG, IgM and IgA levels in the uninfected and recovered subjects.

The presented paper discusses a vitally important issue of the memory B cell behavior regarding anti-SARS-CoV-2 vaccination and the results will undoubtedly contribute to our understanding of the immune response mechanism to SARS-CoV-2 infection.

Minor comments

1.    Line 55: “T3 (three months from the second dose) “  - Is that correct – “second dose”?

2.    Line  131: “between T2 and T3 (3 months before the third dose).” - Is the description in the brackets correct?

Does that match Figure 1? And then, Line 320:  “…three weeks (T2), three months (T3), and six months (T4) after the third booster dose,…”

3.    Check for some typos.

As for general consideration, while comparing the immune response after the third dose of the vaccine in the infected and uninfected subjects, it might be interesting to have a reference cohort of the infected after the second dose only at the same time points.  

Author Response

The manuscript by Rosalia Busà et al. reports the results of a study that evaluated functionality of B cells, including different SARS-CoV-2-specific B cell subpopulations, in a cohort of immunocompetent individuals at three and six months from the third dose of mRNA vaccine. During the follow up period some of the individuals contracted the virus. The authors compared time-related serum Spike-specific IgG, IgM and IgA levels in the uninfected and recovered subjects.

The presented paper discusses a vitally important issue of the memory B cell behaviour regarding anti-SARS-CoV-2 vaccination and the results will undoubtedly contribute to our understanding of the immune response mechanism to SARS-CoV-2 infection.

We thank the reviewer 1 for his/her positive comments on our paper.

 Minor comments

  1. Line 55: “T3 (three months from the second dose) “- Is that correct – “second dose”?

R1. Thanks for the note. You are right, it was a mistake. We now changed “second dose” with “third dose”

  1. Line 131: “between T2 and T3 (3 months before the third dose).” - Is the description in the brackets correct?

Does that match Figure 1? And then, Line 320:  “…three weeks (T2), three months (T3), and six months (T4) after the third booster dose,…”

R2. No, thanks. It was an error, now we correct, in line 131, “before” with “after” the third dose.

  1. Check for some typos.

R3. Thank you very much for your suggestion. We read again carefully the paper to look for any other typos.

As for general consideration, while comparing the immune response after the third dose of the vaccine in the infected and uninfected subjects, it might be interesting to have a reference cohort of the infected after the second dose only at the same time points.

We agree with the reviewer on this intriguing option, but we did not have any SARS-CoV-2 infection in our immunocompetent subjects before the third dose.

Reviewer 2 Report

Good use of statistical analysis to support your study

Author Response

Good use of statistical analysis to support your study

We thanks Reviewer 2 for his/her kind appreciation of our work.

Reviewer 3 Report

The study by R. Busà and colleagues studies the B cell phenotype and the humoral response in a cohort of healthy subjects vaccinated with the BNT162b2 vaccine. The methodologies used to characterize B cells are precise and give new insights to the relationship between B cell differentiation stages and the humoral response against both the vaccine and SARS-CoV-2. Even though the ability to analyze B cells through the experimental design is clear and we think it is the main strongpoint of the work, here we propose some minor suggestions to improve the quality of the work:

1.     The description of the cohort should be improved, since it is not clear the distinction at T4 between recovered and non-recovered subjects. It is stated that for the recovered subjects T4 correspond to both six months after the third dose and simultaneously to one month after the negativization (line 322). Were the patients screened as negative after six months to the third dose included too? If there was a time gap in these cases, it needs to be indicated.

2.     When the characterization of SARS-CoV-2-specific B cells is described (Materials and Methods 4.4) the type of antibody used to analyze the Ig Isotypes (IgM, IgG and IgA) on cell surface needs to be included (type, brand and fluorochrome). In the Appendix B we can see that you used an IgM-APC, an IgA BV-510, and an IgGBV421, we think that they need to be included in the paragraph 4.4 of Materials and Methods and not only in Figure 1 of Appendix B.

3.     Since the cohort is small, it would be suitable to have a control group of non-vaccinated subjects. We understand that it is hard to find non-vaccinated subjects, but as a control for T3 and T4, people who didn’t receive the third dose during the time window of the study could have been included. As an alternative the same analysis could be performed on new people (not previously enrolled) that received the third dose more than six month before the experiments.

4.     In paragraph 2.1 line 97 it is indicated that “at T4 the median value of SARS-CoV-2 specific IgG was significantly higher (p=0.0043)  96 in recovered subjects” but this significancy was not shown in the 1D graph.

5.     In paragraph 2.2 line 130 there is a mistake, since T4 is 3 months after the third dose, and not before.

6.     At line 139 it is indicated that “the neutralizing ability of IgG against the omicron variant is the lowest in comparison to the other VOC”, but the data cited afterwards between brackets seem to be medians of the percentages of neutralization, if so please indicate that.

7.     It is not clear how the data at lines 156 to 160 were depicted in figure 3. It is stated that the increase in the neutralization index regarding the fold increase is significant when comparing recovered to non-recovered subjects, but in the graph the only evident significance is only the one compared to the WT variant. Here we suggest showing two different graphs, depicting the variation of the neutralization index towards the different variants in both recovered and non-recovered subjects, and then eventually indicate the significances between the two graphs.

Author Response

The study by R. Busà and colleagues studies the B cell phenotype and the humoral response in a cohort of healthy subjects vaccinated with the BNT162b2 vaccine. The methodologies used to characterize B cells are precise and give new insights to the relationship between B cell differentiation stages and the humoral response against both the vaccine and SARS-CoV-2. Even though the ability to analyse B cells through the experimental design is clear and we think it is the main strongpoint of the work, here we propose some minor suggestions to improve the quality of the work:

  1. The description of the cohort should be improved, since it is not clear the distinction at T4 between recovered and non-recovered subjects. It is stated that for the recovered subjects T4 correspond to both six months after the third dose and simultaneously to one month after the negativization (line 322). Were the patients screened as negative after six months to the third dose included too? If there was a time gap in these cases, it needs to be indicated.

R1.Thanks for your request for clarification. The subjects recruited in the study are a group of healthy healthcare workers that shared the same vaccination history, similar lifestyle, and who spent a lot of time in close contact in the work environment. We added a graphical abstract in figure 1A to help the understanding of the follow-up. Nobody of the subjects studied was non-recovered, the two groups had the same history until 6 months before the third dose. Six of them remained negative at timepoint T4 while five of them became infected and subsequently negativized, between the fourth and fifth month after the third dose so the period from negativization corresponds to approximately six months from the third dose. In Material and Methods (4.1) we rephrase “In recovered subjects, the time point T4 corresponds to six months from the third dose and simultaneously to one month after negativization” in: “The blood sampling from subjects who contracted the disease was done one month after the negativization, which corresponds to approximately six months from the third dose (T4)”. Concerning the second point, yes, the volunteers were periodically screened through nasopharyngeal swabs and ELISPOT and serology for the Nucleocapsid (N) protein.

  1. When the characterization of SARS-CoV-2-specific B cells is described (Materials and Methods 4.4) the type of antibody used to analyse the Ig Isotypes (IgM, IgG and IgA) on cell surface needs to be included (type, brand and fluorochrome). In the Appendix B we can see that you used an IgM-APC, an IgA BV-510, and an IgGBV421, we think that they need to be included in the paragraph 4.4 of Materials and Methods and not only in Figure 1 of Appendix B.

 R2.Thank you very much for your note. We did not include previously the type of antibody used to analyse the Ig Isotypes because they are part of the Miltenyi kit mentioned, but to avoid misunderstandings, but as you suggest now we added the brand and fluorochrome of Ig Isotypes in Materials and Methods 4.4: “The isolated fraction was stained with SARS-CoV-2 Spike B Cell Analysis Kit (Miltenyi Biotec, Auburn, CA, USA), following the manufacturer’s instructions, and with anti-human CD19BV786,CD27BV605, CD38PE-CF594 and IgDFITC (Becton Dickinson, San Jose, CA, USA), IgMAPC, IgGBV421 and IgABV510 (Miltenyi Biotec, Auburn, CA, USA) monoclonal antibodies, to quantify the SARS-CoV-2-specific B cells, and their related subpopulations, at T4”.

  1. Since the cohort is small, it would be suitable to have a control group of non-vaccinated subjects. We understand that it is hard to find non-vaccinated subjects, but as a control for T3 and T4, people who didn’t receive the third dose during the time window of the study could have been included. As an alternative the same analysis could be performed on new people (not previously enrolled) that received the third dose more than six month before the experiments.

R3.Thanks for your valuable tip, but we don't have any of the suggested control groups. First of all, because any other subjects to be included as a control, being part of the same hospital structure, all received the vaccines in the same period. The choice of these 11 subjects for the study was made precisely because they were followed constantly starting from the second dose and then for the entire follow-up period. Furthermore, the aim of the study was to evaluate the long-term ability of specific memory B cells to respond to the antigen either in vitro (FluoroSpot) or in vivo (natural infection).

4. In paragraph 2.1 line 97 it is indicated that “at T4 the median value of SARS-CoV-2 specific IgG was significantly higher (p=0.0043) 96 in recovered subjects” but this significancy was not shown in the 1D graph

R4. Thanks for your suggestion, we checked it and add the statistical significance in figure 1 (1D graph).

  1. In paragraph 2.2 line 130 there is a mistake, since T4 is 3 months after the third dose, and not

before.

R5.Thank you very much. It was an error, indeed we now correct “before” with “after” the third dose.

  1. At line 139 it is indicated that “the neutralizing ability of IgG against the omicron variant is the lowest in comparison to the other VOC”, but the data cited afterwards between brackets seem to be medians of the percentages of neutralization, if so please indicate that.

R6. Thanks for your valuable tip, we rephrase this part in “To note that, at both time points, the median of the percentages of neutralization of IgG against the omicron variant is the lowest in comparison to the other VOC, as a matter of fact, at T2 the median of the percentage of neutralization was 41.38% (SEM=10.24), that decrease at 38.16% at T4 (SEM=9.25) in only vaccinated subjects (Figure 2F).

  1. It is not clear how the data at lines 156 to 160 were depicted in figure 3. It is stated that the increase in the neutralization index regarding the fold increase is significant when comparing recovered to non-recovered subjects, but in the graph the only evident significance is only the one compared to the WT variant. Here we suggest showing two different graphs, depicting the variation of the neutralization index towards the different variants in both recovered and non-recovered subjects, and then eventually indicate the significances between the two graphs.

R7.Thank you very much for your suggestion. We added two different graphs showing the fold change (decrease) in each group, but we leave the previous graph in which we showed the comparative analysis for both groups together. We discuss these changes in results (2.3): “Interestingly, normalizing the SARS-CoV-2-specific IgG neutralization VOC against the wild-type, we observed a general reduction in neutralization ability versus all VOC in never infected subjects (Figure 2A), particularly we find significantly decrease in gamma (p= 0.0332) and omicron (p<0.0001) VOC. While, concerning the recovered subjects (Figure 2B) we observed only a slightly and not significant reduction in omicron VOC. At last, to better quantify the increased index in neutralization ability in recovered subjects, we compared the fold change of both groups (Figure 3C), finding a significant increase in neutralization ability in recovered subjects compared to uninfected, against beta (p=0.0363), gamma (p=0.0057), and omicron (p<0.0001) variants, in which we observed a 2-fold increase”, and in the legend to figure 3: “Figure 3. Fold change in SARS-CoV-2-specific IgG neutralization against different VOC (alfa UK, beta, gamma, delta, and omicron) normalized versus the wild-type (WT) in uninfected (A) and recovered (B) vaccinated subjects at T4. (C) Comparison in fold change between the two analysed groups at T4. . The significance was determined using Dunnett’s multiple comparisons test: *p<0.0332, **p<0.0021, ****p < 0.0001”.